# A hidden battle in the dirt: Soil amoebae interactions with *Paracoccidioides* spp

Patrícia Albuquerque[1,2,3]*, André Moraes Nicola[3,4,5], Diogo Almeida Gomes Magnabosco[2], Lorena da Silveira Derengowski[2,6], Luana Soares Crisóstomo[2], Luciano Costa Gomes Xavier[2], Stefânia de Oliveira Frazão[2], Fernanda Guilhelmelli[2], Marco Antônio de Oliveira[2], Jhones do Nascimento Dias[2], Fabián Andrés Hurtado[2], Marcus de Melo Teixeira[4], Allan Jefferson Guimarães[7], Hugo Costa Paes[4], Eduardo Bagagli[8], Maria Sueli Soares Felipe[2,3], Arturo Casadevall[3], Ildinete Silva-Pereira[2]

**1** Faculty of Ceilândia, University of Brasília, Brasília, DF, Brazil, **2** Department of Cell Biology, Institute of Biological Sciences, University of Brasília, Brasília, DF, Brazil, **3** Department of Molecular Microbiology and Immunology, Johns Hopkins Bloomberg School of Public Health, Baltimore, MD, United States of America, **4** Faculty of Medicine, University of Brasília, Brasília, Brazil, **5** Program in Genomic Sciences, Catholic University of Brasília, Brasília, DF, Brazil, **6** Military College, Brasília, Brazil, **7** Department of Microbiology and Parasitology, Biomedical Institute, Fluminense Federal University, Niteroi, RJ, Brazil, **8** Department of Microbiology and Immunology, Biosciences Institute, UNESP, Botucatu, SP, Brazil

* palbuquerque@unb.br, patricia.andrade@phd.einstein.yu.edu

**Data Availability Statement:** All relevant data are within the manuscript and its Supporting Information files.

## Abstract

*Paracoccidioides* spp. are thermodimorphic fungi that cause a neglected tropical disease (paracoccidioidomycosis) that is endemic to Latin America. These fungi inhabit the soil, where they live as saprophytes with no need for a mammalian host to complete their life cycle. Despite this, they developed sophisticated virulence attributes allowing them not only to survive in host tissues but also to cause disease. A hypothesis for selective pressures driving the emergence or maintenance of virulence of soil fungi is their interaction with soil predators such as amoebae and helminths. We evaluated the presence of environmental amoeboid predators in soil from armadillo burrows where *Paracoccidioides* had been previously detected and tested if the interaction of *Paracoccidioides* with amoebae selects for fungi with increased virulence. Nematodes, ciliates, and amoebae–all potential predators of fungi–grew in cultures from soil samples. Microscopical observation and ITS sequencing identified the amoebae as *Acanthamoeba* spp, *Allovahlkampfia spelaea*, and *Vermamoeba vermiformis*. These three amoebae efficiently ingested, killed and digested *Paracoccidioides* spp. yeast cells, as did laboratory adapted axenic *Acanthamoeba castellanii*. Sequential co-cultivation of *Paracoccidioides* with *A. castellanii* selected for phenotypical traits related to the survival of the fungus within a natural predator as well as in murine macrophages and in vivo (*Galleria mellonella* and mice). These changes in virulence were linked to the accumulation of cell wall alpha-glucans, polysaccharides that mask recognition of fungal molecular patterns by host pattern recognition receptors. Altogether, our results indicate that *Paracoccidioides* inhabits a complex environment with multiple amoeboid predators that can exert selective pressure to guide the evolution of virulence traits.

**Funding:** This work was supported by grants from the Brazilian agencies, Conselho Nacional de Desenvolvimento Científico e Tecnológico (CNPq) (http://www.cnpq.br/) and Fundação de Apoio à Pesquisa do Distrito Federal (FAP-DF-Brazil) (www.fap.df.gov.br/) to PA and ISP. AC was supported in part by NIH grants 5R01HL059842, 5R01AI052733. This study was also partially financed by scholarships from Coordenação de Aperfeiçoamento de Pessoal de Nível Superior (Capes-Brazil, Finance code 001). The funders had no role in study design, data collection and analysis, decision to publish, or preparation of the manuscript.

**Competing interests:** The authors have declared that no competing interests exist.

## Author summary

Fungi from the genus *Paracoccidioides* cause paracoccidioidomycosis, a neglected tropical disease that mainly affects poor rural workers in Latin America. *Paracoccidioides* can live its whole life in the soil, without the need to infect humans or other animals to complete its life cycle. Studies with other such free-living organisms suggest they have acquired the ability to survive and cause disease in humans by interacting with soil predators such as amoebae and worms. In this study we have investigated organisms present in soil in which *Paracoccidioides* had been previously detected, with a focus on predatory amoebae. Cultures from soil samples showed numerous amoebae, which were isolated and identified using genetic tools. These amoebae were able to ingest and destroy *Paracoccidioides* yeast cells. More detailed experiments made with a laboratory strain of *Acanthamoeba castellanii* showed that the interaction with amoebae did increase the fungal ability to survive in and kill not only cells such as amoebae and mouse macrophages, but also whole organisms such as *Galleria mellonella* larvae and mice. This better understanding of the habitat in which *Paracoccidioides* lives in nature might lead to new and improved strategies to prevent infection and thus mitigate the burden of this disease.

## Introduction

Human beings are constantly challenged by microorganisms in virtually every environment and circumstance. Effective host immune responses, however, ensure that very few of them cause disease. Pathogenic microorganisms usually have a complex set of virulence attributes that allow them to evade immune effectors, proliferate and cause diseases [1]. Immunity is a crucial selective pressure driving the evolution of virulence attributes in microbial pathogens tightly associated with mammalian hosts. However, the evolution of virulence in microbes that do not need to interact with mammals to complete their life cycles, such as the agents of most fungal invasive diseases, is far less clear. These agents include pathogenic species in the genus *Paracoccidioides*. Five species in this genus of thermodimorphic fungi, *P. brasiliensis*, *P. americana*, *P. restrepiensis*, *P. venezuelensis* and *P. lutzii*, cause paracoccidioidomycosis (PCM), one of the most prevalent systemic mycoses in Latin America [2, 3]. This neglected disease is an important cause of morbidity and mortality among men from rural areas in these countries. Infection occurs by the inhalation of airborne fungal propagules (mycelium fragments or conidia) from the environment, and most infections are asymptomatic. However, some patients do develop PCM, which ranges from mild pneumonia to life-threatening systemic disease [4, 5].

In the last two decades, a number of studies done with other species of invasive fungal pathogens (*Cryptococcus neoformans*, *C. gattii*, *Sporothrix schenckii*, *Blastomyces dermatitidis*, *Histoplasma capsulatum*, and *Aspergillus fumigatus*) have provided a compelling explanation for the evolution of virulence in these soil saprophytes: avoiding predation by soil amoebae requires phenotypical traits that also provide protection against mammalian immune defenses and are thus associated with virulence [6–10]. Each of these fungi survived after co-cultivation with phagocytic unicellular organisms such as *Acanthamoeba castellanii* and *Dictyostelium discoideum* due to phenotypical traits that are also effective in evading human macrophages. Moreover, their co-cultivation with amoebae selects for survivors that are more virulent in mammalian models. Exposure to other soil predators such as ciliates and helminths suggests a more complex interaction scenario, beyond those seen with amoebae [11, 12]. These studies,

however, were performed in controlled laboratory conditions using mostly pure and axenic cultures of laboratory-adapted predators; this very informative system is nonetheless an extreme simplification of the complex ecosystem soil saprophytes find in nature. In this work, we have delved further into the ecology of the soil environment in a region where *Paracoccidioides* spp. had previously been confirmed by nested PCR, studying both the composition of predator populations and the interaction between some of these and *Paracoccidioides* cells.

## Materials and methods

### *Paracoccidioides* spp. maintenance and preparation for interaction

For our studies we used three *Paracoccidioides* spp. isolates: Pb18, a *P. brasiliensis* clinical isolate; PbT16B1, a *P. brasiliensis* isolate that has been previously obtained from the spleen of a nine-banded armadillo (*Dasypus novemcinctus*) [13] and Pb01, a *P. lutzii* clinical isolate. PbT16B1 was obtained from the mycology collections of the Fungal Biology Laboratory (Department of Microbiology and Immunology, Biosciences Institute, UNESP, Botucatu, SP). Pb18 and Pb01 are part of the mycology collection of the University of Brasilia (Department of Cell Biology). The yeast phase of these isolates was maintained by subculturing every seven days in Fava-Netto's medium (1% w/v peptone, 0.5% w/v yeast extract 0.3% w/v proteose peptone, 0.5% w/v beef extract, 0.5% w/v NaCl, 4% w/v glucose, and 1.4% w/v agar, pH 7.2) or GPY medium (2% glucose w/v, 1% peptone w/v, 0.5% yeast extract w/v and 2% agar w/v) and incubating at 37°C. For experiments, five-day cultures were used. Before interaction assays, the fungal cells were collected, washed three times with PBS and diluted to the appropriate cell densities. Only cultures with viability above 80%, as measured with the viability dye phloxine B (Sigma-Aldrich), were used.

### Axenic amoebae

*A. castellanii* 30234 (*American Type Culture Collection*—ATCC, Manassas, VA, USA) was cultivated in PYG medium (2% proteose peptone, 0.1% yeast extract, 1.8% glucose, 0.1% sodium citrate dihydrate, 2.5 mM $Na_2HPO_4$, 2.5 mM $KH_2PO_4$, 4 mM $MgSO_4$, 400 μM $CaCl_2$ and 50 μM $Fe(NH_4)_2(SO_4)_2$) at 28°C as previously described [8].

### Soil amoeba isolation and maintenance

Soil amoebae were isolated from samples of armadillo burrows located at Lageado Farm (−22° 50' 14.36" latitude and −48° 25' 31.35″longitude), an area where the armadillo from which PbT16B1 was isolated was captured; in this location the fungus had been also detected in soil by nested PCR [13]. Additionally, rural workers that have lived and/or worked in this region were diagnosed with or died from PCM [14]. About five grams of each soil sample were mixed with 20 mL of sterile Page's modified Neff's amoeba saline (PAS– 2 mM NaCl, 33 mM $MgSO_4$, 27 mM $CaCl_2$, 1 mM $Na_2HPO_4$, 1 mM $KH_2PO_4$) and vigorously mixed to homogenize the samples. After sedimentation for 5 minutes, 100 μL of each sample were spread over a plate of non-nutrient agar (PAS + 1.5% agar) containing a lawn of heat-killed *Escherichia coli* OP50. The plates were incubated at 25°C for 10–14 days and observed daily by light microscopy for the presence of amoeba cysts or trophozoites [15, 16]. Agar sections containing cysts or trophozoites were cut and transferred to new plates to enrich the cultures. Finally, amoebae were transferred to PAS, counted and submitted to limiting dilution cloning. It was not possible to obtain axenic cultures, so these freshly isolated amoebae were maintained in PAS or in non-nutrient agar plates with *E. coli* strain OP50 as a food source.

### DNA isolation for typing of soil amoebae

DNA extractions from soil amoebae were performed using the UNSET protocol [17] or QIAamp DNA Blood Mini Kit (Qiagen). Identification of amoeba isolates was performed by PCR using common amoeba primers AmeF977 (GATYAGATACCGTCGTAGTC) and AmeR1534 (TCTAAGRGCATCACAGACCTG) [18] or *Acanthamoeba* specific primers JDP1 (GGCCCAGATCGTTTACCGTGAA) and JDP2 (TCTCACAAGCTGCTAGGGAGTCA) [19]. PCR fragments were purified and cloned into the TOPO TA vector (Thermo Fisher) and transformed into DH5-α *E. coli*. At least three plasmid clones were purified and Sanger-sequenced for each amoeba isolate. Sequences were blasted against GenBank and deposited under BioProject 506281. All the sequences for clones derived from each one of the 7 isolates resulted in the same BLAST hits.

### Soil amoeba and *P. brasiliensis* interaction assays

The distinct amoeba isolates were collected from our culture stocks, washed three times to remove bacteria, plated onto glass-bottom plates and co-incubated with *P. brasiliensis* cells previously dyed with FITC or pHrodo (Thermo Fisher). The multiplicity of infection (MOI) was of one and co-incubation was carried out for 24 hours at 25°C. The samples were then dyed with Uvitex 2B (Polysciences, Inc) to distinguish intracellular and extracellular fungal cells [20] and observed in a Zeiss Axio Observer Z1 inverted microscope using a 40X/NA 0.6 objective for quantification of phagocytosis. A minimum of 100 amoebae per sample was analyzed, and the experiments were performed at least three times on different days. Alternatively, predation assays in which soil amoebae were incubated in solid non-nutrient agar with a lawn of *P. brasiliensis* cells were performed as described below. Soil amoeba viability after the interaction was assessed by Trypan blue exclusion as previously described [8].

### Soil amoeba and *P. brasiliensis* predation assays

*P. brasiliensis* yeast cells were washed in PBS and big cell clumps were removed by passing the cells through 40-micron cell strainers or by multiple passages through 26-Gauge needles. After that, cell density was determined with a hemocytometer and a specific number of fungal cells were plated in the center of non-nutrient agar plates. After the fungal cell lawn dried, we aliquoted a suspension of the different soil amoeba isolates in the center of the plates. The plates were kept at 25°C and examined daily for the presence of fungal cell lysis plaques.

### Phagocytosis and killing assay for the interaction of *Paracoccidioides* spp with axenic *A. castellanii*

Cells of *A. castellanii* were plated onto 96- or 24-well microplates at $5 \times 10^4$ and $2 \times 10^5$ cells/well, respectively, and incubated with yeast cells ($1 \times 10^5$ and $4 \times 10^5$ cells/well) for different time intervals (6, 24 or 48 hours) at 28°C or 37°C (MOI = 2). After co-incubation, the supernatant was discarded, the cells were fixed with cold methanol for 30 min at 4°C and overnight stained with Giemsa. The samples were then observed and photographed in a Zeiss Axio Observer Z1 inverted microscope. At each condition, the percentage of phagocytosis was evaluated after Giemsa staining. Alternatively, phagocytosis was evaluated by fluorescence microscopy using fungal cells previously dyed with CMFDA or FITC before the interaction. A minimum of 100 amoebae per sample was analyzed, and the experiments were performed at least three times on different days. *A. castellanii* viability after the interaction was assessed by Trypan blue exclusion as previously described [8]. Fungal survival after the interaction was assessed by CFU counting after amoeba lysis as described below.

## *Paracoccidioides* spp. survival after interaction with amoebae

To assess survival of the fungus upon co-incubation with amoebae, the organisms were co-cultured for six or 24 hours in 24-well plates (MOI of two). The cells were then detached from the plates and submitted to 5–8 passages through a 26-gauge syringe to lyse the amoebae. The remaining yeast cells were serially diluted and plated onto solid BHI supplemented with 4% horse serum, 5% conditioned medium of the Pb192 strain of *P. brasiliensis* (BHI-sup) [21] and chloramphenicol (34 μg/mL). The plates were incubated at 37˚C for 7–10 days for colony counting. For each condition at least three wells were analyzed, and the experiments were performed at least three times on different days.

## Confocal microscopy

Sterile cover glasses were placed on six-well plates and $5 \times 10^6$ amoeba cells in PYG medium were added to each well. After two hours of adhesion, amoebae were labelled with fluorescent dyes (DiD-DS for cell membrane or DDAO-SE for intracellular proteins). After that the amoebae were co-incubated with $10^7$ cells of *P. brasiliensis* (MOI of two) previously dyed with CMFDA. After two hours of interaction, the cover glasses were washed, fixed with cold methanol and mounted onto slides for confocal microscopy in a Leica SP5 microscope using a 63x NA 1.4 objective.

## Transmission electron microscopy (TEM)

For electron microscopy, *P. brasiliensis* yeast cells were incubated alone or in the presence of amoebae (MOI of two) for two, four or 24 hours at 25˚C or 28˚C. Samples were fixed in 2.5% glutaraldehyde, 3 mM $MgCl_2$, 0.1 M sodium cacodylate buffer, pH 7.2 overnight at 4˚C. After rinsing with buffer, samples were post fixed in 1% osmium tetroxide in buffer (1 hour) on ice in the dark followed by another rinse with 0.1 M sodium cacodylate buffer. Samples were left at 4˚C overnight in buffer, rinsed with 0.1 M maleate buffer, *en bloc* stained with 2% uranyl acetate (0.22 μm filtered, 1 hour, in the dark) in 0.1 M maleate, dehydrated in a graded series of ethanol and propylene oxide, and embedded in Eponate 12 (Ted Pella) resin [22]. Samples were polymerized at 60˚C overnight. Thin sections, 60 to 90 nm each, were cut with a diamond knife on the Reichert-Jung Ultracut E ultramicrotome and picked up with naked 200 mesh copper grids. Grids were stained with 2% uranyl acetate (aq.) followed by lead citrate and observed under a Philips CM120 TEM at 80 kV. Images were captured with an AMT XR80 high-resolution (16-bit) 8 Mpixel camera.

## Scanning electronic microscopy (SEM)

For electron microscopy, *P. brasiliensis* yeast cells were incubated alone or in the presence of amoebae (MOI of two) in PAS for two, four or 24 hours at 25˚C or 28˚C in 24-wells plates containing poly-lysine coverslips. After that, the samples were processed for SEM following previously published protocols with some modifications. Briefly, samples were fixed in 2.5% glutaraldehyde, 3 mM $MgCl_2$, in 0.1 M sodium cacodylate buffer, pH 7.2 overnight at 4˚C. After rinsing with buffer, samples were post fixed in 1% osmium tetroxide in buffer (1 hour) on ice in the dark followed by two distilled water rinses and dehydration in ethanol. [23]. Samples were dried for SEM with HMDS and mounted on carbon coated stubs, coated with 20 nm AuPd and imaged on a Leo FE-SEM at 1 kV.

## Sequential interaction of *A. castellanii* and *Paracoccidioides* spp

We co-cultured *Paracoccidioides* spp. cells with *A. castellanii* for six hours at 28˚C in PYG medium at a MOI of two. The cells were then detached from the plates and passed 5–8 times

through a 26-Gauge syringe to lyse amoebae. The remaining yeast cells were plated onto solid BHI-Sup (4% horse serum, 5% conditioned medium of the Pb192 strain of *P. brasiliensis*, 34 μg/mL chloramphenicol). The plates were incubated at 37˚C for a week, and the recovered cells were collected from the plates, washed three times with PBS, counted and used in a subsequent round of interaction with *A. castellanii* for another six hours. This process was repeated a total of five times and the resulting passaged strains were then named Pb18-Ac. The Pb18 strain, cultured in PYG for six hours at 28˚C and then plated onto BHI-Sup was used for comparison.

### *Galleria mellonella* infection

Wax moth larvae were kept in glass bottles in a dark environment in an incubator at 29˚C. The colony was maintained on an artificial diet consisting of portions of 500 g of Infant Cereal, 100 g of saccharose, 100 mL of glycerin, 100 g of honey and 100 mL of distilled water [24]. Larvae weighting between 180 and 250 mg were used in the survival tests. Prior to each experiment larvae were collected, randomized into groups of 12–16 individuals and surface-cleaned with ethanol 70%. Each group received an injection of 10 μl of PBS or yeast cell suspension (Pb18 or Pb18-Ac) at $10^6$ cells/mL in the hind left proleg. All yeast suspensions contained ampicillin (20 mg/kg) to prevent infection with bacteria from the surface of the larva. The groups of infected larvae were placed in Petri dishes, incubated at 37˚C and daily monitored for survival [25].

### Mouse infection and survival analysis

We infected isogenic 10-week-old BALB/c male mice with Pb18-Ac or the non-passaged Pb18. The cells from each group were collected from BHI-sup plates after five days of culture, washed in PBS, counted, assessed for viability and diluted to the appropriated cell densities. The mice were anesthetized using a combination of 100 mg/kg of body weight ketamine and 10 mg/kg of body weight xylazine administered intraperitoneally. For infection, $10^6$ cells of either sample were intratracheally inoculated into two groups of 14 mice each. The animals were clinically monitored during 12 months after infection and moribund animals (defined by lethargy, dyspnea, and weight loss) were euthanized. The experiment was set up as a blind assay: the investigators who infected and monitored the mice did not know which strain had been administered to each group until after the experiment finished.

### Quantitative RT-PCR of *P. brasiliensis* Pb18 and Pb18-Ac genes potentially involved in host-pathogen interaction

Total RNA of Pb18 and Pb18-Ac cells was extracted using the Qiagen RNeasy Plant Minikit according to the supplied protocol, including treatment with the supplier's RNAse-free DNAse to avoid contamination with DNA. Samples were quantified in a Nanodrop spectrophotometer (Thermo Fisher). For cDNA synthesis, 2 μg of total RNA from each sample were reverse-transcribed using the High-Capacity cDNA Reverse Transcription Kit (Thermo Fisher) according to the manufacturer's instructions. The cDNAs were used as template for qPCR in triplicate using the Fast SYBR Green Master Mix (Thermo Fisher) with cycling conditions according to manufacturer's protocol, adapted for a reaction volume of 10 μL. The L34 transcript was used as endogenous control [26]. The primers used had their amplification efficiency assessed by the standard curve method and are listed on S1 Table. Changes in transcript abundance were quantified by the $2^{-\Delta\Delta Ct}$ method [27], with fold-change determined as the ratio of values for each transcript between the non-passaged and passaged strains. The oligonucleotides used in these experiments are described in S1 Table.

### Flow cytometry analysis for the detection of α-glucan at the fungal cell wall

After the fifth cycle of interaction with amoeba Pb18-Ac and Pb18 cells were plated on solid BHI-sup and grown for about a week. After that, the cells were collected, washed 3 times with PBS, paraformaldehyde-fixed and incubated with MOPC 104E (Sigma), an antibody that specifically recognizes α-(1,3)-glucan. They were then washed and incubated with a secondary antibody to IgM conjugated with Alexa Fluor 488 (Thermo Fisher). After washing, cell suspensions were analyzed in a BD LSR Fortessa flow cytometer. The resulting data were analyzed using FlowJo software.

### Statistical analyses

All statistical analyses were performed using GraphPad Prism 8.0 (GraphPad software). Percentage phagocytosis and percentage of amoeba viability (% Dead amoeba) were evaluated using Fisher's exact test. Survival curves were analyzed using log-rank and Wilcoxon tests. For CFU experiments we used one-way ANOVA with Tukey's multiple comparison post-test or unpaired t-test when comparing only two samples. The quantitative PCR analysis was performed with unpaired t-tests.

### Ethics statement

All mouse experiments were pre-approved by the Committee for Use of Animals in Research of the Catholic University of Brasília (protocol 017/14) in agreement with Brazilian laws for use of experimental animals and the Ethical Principles in Animal Research adopted by the Brazilian College for Control of Animal Experimentation.

## Results

### Multiple groups of potential predators are present in the environment in which *P. brasiliensis* lives

Initial microscopical analysis of cultures obtained from soil samples positive for *Paracoccidioides* DNA as schematically represented in Fig 1A revealed the presence of multiple potential predators, including several amoeba morphotypes, ciliates, and nematodes (Fig 1B–1I). Although ciliates and nematodes are known *C. neoformans* predators [11, 12], we chose to focus on ameboid predators. After using limiting dilution to obtain plates that seemed to contain only one type of amoeba, we made several attempts to establish axenic or monoxenic cultures. We were not successful, however, even in the presence of several antibiotics. We purified DNA from the different isolates and performed PCR using primers specific to *Amoebozoa* and for *Acanthamoeba* spp identification. Sequencing and comparison against GenBank revealed that we had two *Allovahlkampfia spelaea* isolates, three *Vermamoeba vermiformis* (formerly *Hartmannella vermiformis*) isolates and two *Acanthamoeba* spp isolates (Sequences were deposited under BioProject 506281). All sequences from the same isolates resulted in the same hits at GenBank.

### Soil amoeba isolates interact with and kill *P. brasiliensis* yeast cells

We tested if these soil amoebae were able to phagocytose *P. brasiliensis* cells by co-incubating them for 24 hours in PAS after adding antibiotic and removing most of the bacterial cells that were used to feed the amoebae. The three amoeba isolates were each able to phagocytose *P. brasiliensis* cells (Fig 2A), even in the presence of remaining bacterial cells from amoeba cultures, which are probably a preferential food source. We also observed that the isolated

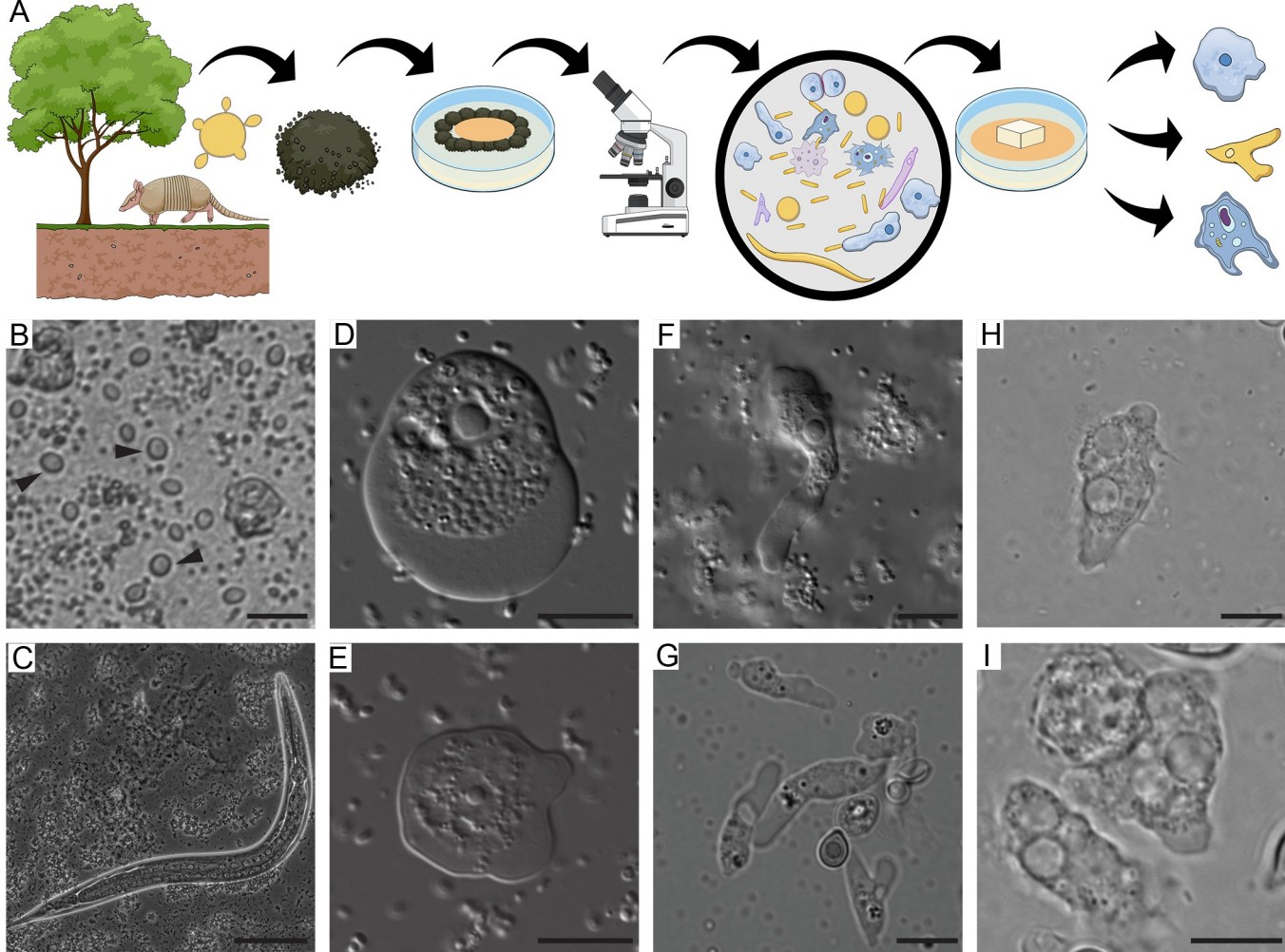

**Fig 1. Soil organisms sharing the putative habitat of *P. brasiliensis*.** A) Schematic representation of the soil amoeba isolation methodology. Soil samples from armadillo burrows positive for *P. brasiliensis* DNA were collected and used for the isolation of soil amoebae. The samples were plated in non-nutrient agar plates containing a bacterial lawn as a food source and observed in an inverted microscope B) Bright field microscopy of ciliate trophozoites (black arrowheads) present in a soil sample. Scale bar = 20 **μ**m, C) Bright field microscopy of a nematode present in the soil sample. Scale bar = 50 **μ**m, D) and E) DIC microscopy of trophozoites of *A. spelaea*. Scale bar = 10 **μ**m. F and G) DIC microscopy of trophozoites of *V. vermiformis*. Scale bar = 10 **μ**m. H and I) DIC microscopy of trophozoites of *Acanthamoeba* spp. Scale bar = 10 **μ**m.

*Acanthamoeba* spp. had decreased viability after 24 hours of co-incubation with *P. brasiliensis*, while the isolated *A. spelaea* was able to survive better in the presence of yeast cells (Fig 2B).

The presence of antibiotic-resistant bacteria in the isolated amoebae cultures prevented us from evaluating *P. brasiliensis* viability after interaction with soil amoebae by CFU counting. Due to the slow growth rate of this fungus, all plates were covered with bacteria before we could observe fungal colonies. To address this limitation, we performed predation assays in non-nutrient agar plate where *P. brasiliensis* lawns were confronted with soil amoeba isolates. We were able to observe regions of fungal cell clearance starting at 7 days of interaction with all the amoeba isolates as exemplified by the agar plate containing *P. brasiliensis* cells with *A. spelaea* shown in Fig 2C–2F. After seven days of co-incubation, we could see many trophozoites mixed in the fungal cell lawn and around the colony (Fig 2D). Most fungal cells had altered morphology, resembling dead empty shells (Fig 2E), and we observed some fungal cells interacting with amoebae (Fig 2F). Interactions with the other isolates are depicted in S1 Fig.

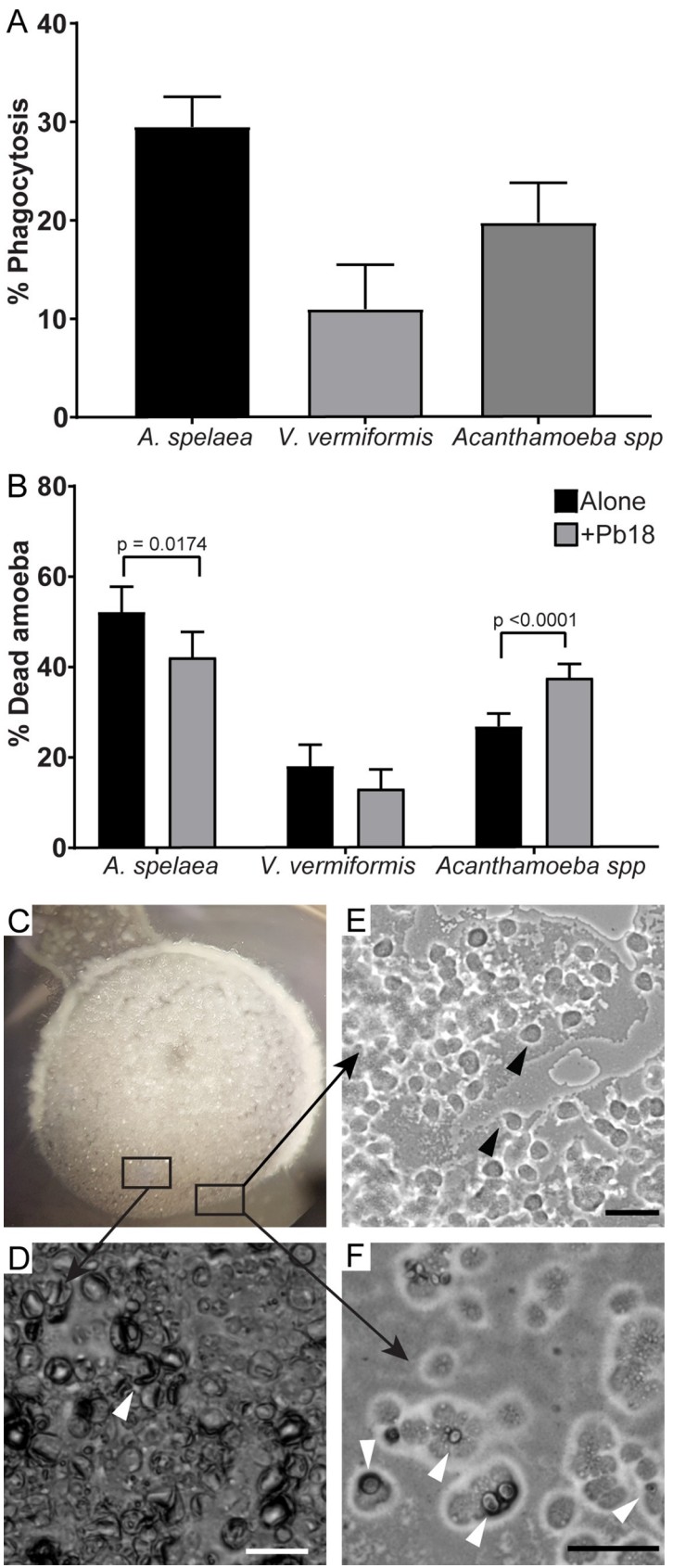

**Fig 2. Interaction between *P. brasiliensis* Pb18 with amoebae isolated from soil of armadillo burrows positive for *P. brasiliensis*.** The amoeba isolates were co-incubated with Pb18 previously dyed with pHrodo or FITC at an MOI of two at 25°C for 24 hours in PYG medium. A) Percentage of amoeba cells interacting with *P. brasiliensis* Pb18. After the interaction, non-internalized Pb18 cells were dyed using Uvitex 2B. B) Viability of the different amoeba isolates after 24 hours of interaction with Pb18. A and B depict the results of at least three independent experiments. At least 100 cells per replicate of each sample were counted for each assay. The bars represent 95% confidence intervals. C-F) a suspension of *A. spelaea* cells was placed next to a colony of *P. brasiliensis* cells in non-nutrient agar. The cells were co-incubated at 25°C and examined daily in an inverted microscope. C) Macroscopic view of the fungal colony in a 35 mm plate. D) Microscopic view of the fungal cell lawn after seven days of interaction. E) Microscopic view of amoeba trophozoites growing in the periphery of the fungal lawn. F) Microscopic view of amoeba trophozoites interacting with a fungal cell. Scale bars = 50 μm. Black arrowheads indicate trophozoites. White arrowheads indicate fungal cells.

Additionally, after two weeks or more of interaction, we observed scarce fungal filamentation in the co-culture samples, possibly because most fungal cells were dead, while the control fungal spots without amoebae displayed intense filamentation (S2 Fig).

We further evaluated the fungal interaction with three different species of amoeba in saline suspension after 4 or 24 hours of interaction by TEM and SEM as presented in Fig 3. TEM analysis revealed internalized fungal cells and/or cell wall debris inside amoeba vacuoles of the three different species, *A. spelaea* (Fig 3A and 3B), *Acanthamoeba* spp (Fig 3E and 3F) and *V. vermiformis* (Fig 3I and 3J) at both time points. SEM confirmed the contact between the three different amoeba species with *P. brasiliensis* in all the interactions (Fig 3C, 3D, 3G, 3H, 3K and 3L). It should be noted that *V. vermiformis* cells can be considerably smaller than large *P. brasiliensis* mother cells (Fig 3K and 3L). Both electron microscopy approaches revealed extreme morphological alterations in most fungal cells upon interaction with the three amoebae in comparison with fungal cells growing alone, which were included as a control (S3 Fig, panels A-D). Control fungal cells displayed more preserved cytoplasm contents and overall morphology in both TEM and SEM. On the other hand, we observed many collapsed fungal cells in the co-cultures, including mother cells with shrinking buds and perforations in the cell wall that might explain the observation of empty cell wall shells in TEM (Fig 3 and S3 Fig). Altogether these results confirm the ability of amoebae to kill *Paracoccidioides* and suggest that strategies beyond phagocytosis of fungi must be considered to explain how they do it.

## *Acanthamoeba castellanii* from axenic cultures can efficiently phagocytose and kill *P. brasiliensis* cells

Since the soil bacteria that remained in amoeba cultures were a third component of the microbial interaction system, and therefore a confounding factor, we decided to further evaluate the interaction of fungal cells with soil amoebae using axenized cultures of *A. castellanii*. Analysis of the co-culture of *A. castellanii* with Pb18 by light microscopy after Giemsa staining (Fig 4A and 4B), transmission electron microscopy (Fig 4C) and confocal microscopy (Fig 4D) revealed the interaction with and ingestion of yeast cells by amoebae. The cell wall-labeling dye Uvitex 2B (blue), which does not penetrate cells that are viable or not permeabilized, confirmed that some fungi were internalized. The black arrow in Fig 4D indicates an internalized yeast cell that is not labeled with CMFDA, which together with the irregular morphology suggests that this yeast cell is probably dead.

The percentage of phagocytosis of *P. brasiliensis* by *A. castellanii* was followed at different time intervals, from 30 minutes to 24 hours of interaction. It varied from 39% at 30 minutes to 68% at six hours (S4 Fig).

We also evaluated the outcome of amoeba predation by measuring fungal cell viability by CFU after six and 24 hours of interaction with *A. castellanii* cells. There was no significant reduction in fungal survival with or without amoeba at the earlier time point (Fig 4E), but the

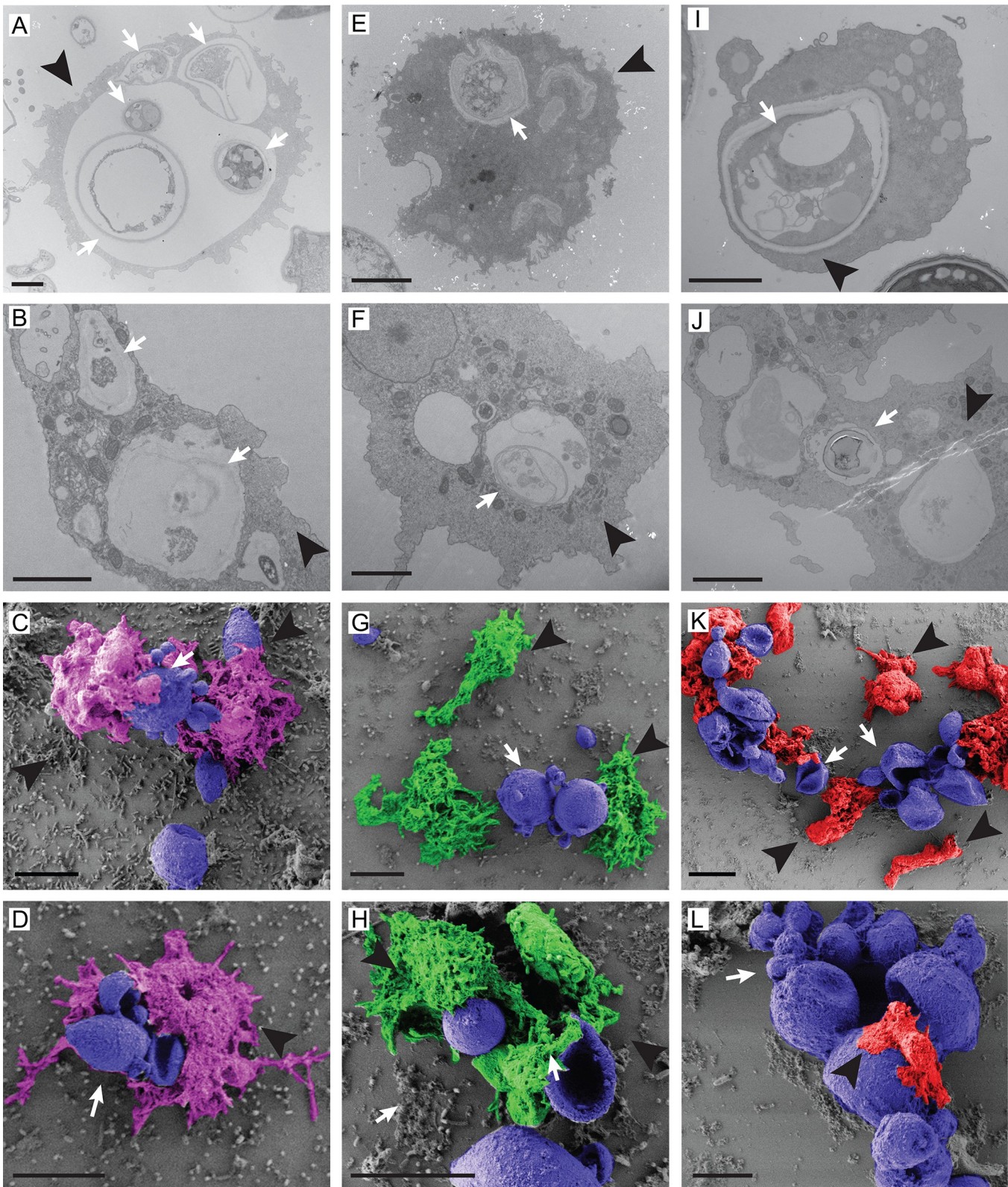

**Fig 3. TEM and SEM analysis of the interaction of *P. brasiliensis* Pb18 cells and different soil amoebae.** The isolates were co-incubated with Pb18 at an MOI of two at 25°C for 4 or 24 hours in PAS and fixed for microscopy. A-B) TEM of the interaction of *P. brasiliensis* with *A. spelaea* (4 h, 24 h). Scale bars = 500 nm. C-D) SEM of the interaction of *P. brasiliensis* with *A. spelaea*. Scale bars = 10 μm. E-F) TEM of the interaction of *P. brasiliensis* with *Acanthamoeba spp* (4 h, 24

h). Scale bars = 2 μm. G-H) SEM of the Interaction of *P. brasiliensis* with *Acanthamoeba spp.* Scale bars = 10 μm. I-J) TEM of the interaction of *P. brasiliensis* with *V. vermiformis* (4 h, 24 h). Scale bars = 2 μm. K-L) SEM of the interaction of *P. brasiliensis* with *V. vermiformis.* Scale bars = 10 μm. White arrows indicate fungal cells, or their remains and black arrowheads indicate amoeba cells.

number of fungal CFUs was reduced by 90% after 24 hours of interaction (Fig 4F), indicating that *A. castellanii* was very efficient in fungal killing. On the other hand, the trypan blue exclusion assay on amoebae after interaction with *P. brasiliensis* showed that amoeba viability was barely affected by the fungus. We only found a small difference in their viability at the six-hour time point at 28˚C, but not at other times points at this temperature or at any time points at 37˚C when compared to the non-infected controls (S5A Fig). To evaluate whether this effect resulted from a broader loss of virulence due to in vitro subculturing of the fungus, we tested the virulence of the same Pb18 isolate against J774 macrophages. In contrast with our observations with amoebae, we observed a significant decrease in macrophage viability after 24 (a 16% to 25% increase in dead cells) or 48 hours of interaction (22% to 33%) (S5B Fig).

### There are differences in the ability of different strains of *Paracoccidioides* spp. to survive interaction with amoebae

We also evaluated the interaction of *A. castellanii* with *P. lutzii* (Pb01) and *P. brasiliensis* PbT16B1, an isolate obtained from an armadillo. *A. castellanii* was able to internalize cells of the three different isolates of *Paracoccidioides* spp. at similar rates (Fig 5A). There was no difference in the ability of *P. lutzii* strain Pb01 relative to Pb18 to kill amoebae or to survive at six hours of interaction (Fig 5B and 5C). However, co-incubation with T16B1 resulted in a time-dependent increase in the amoeba mortality in comparison to both other strains (Fig 5B and S6 Fig), while the other two isolates were able to induce a transient decrease in amoeba viability

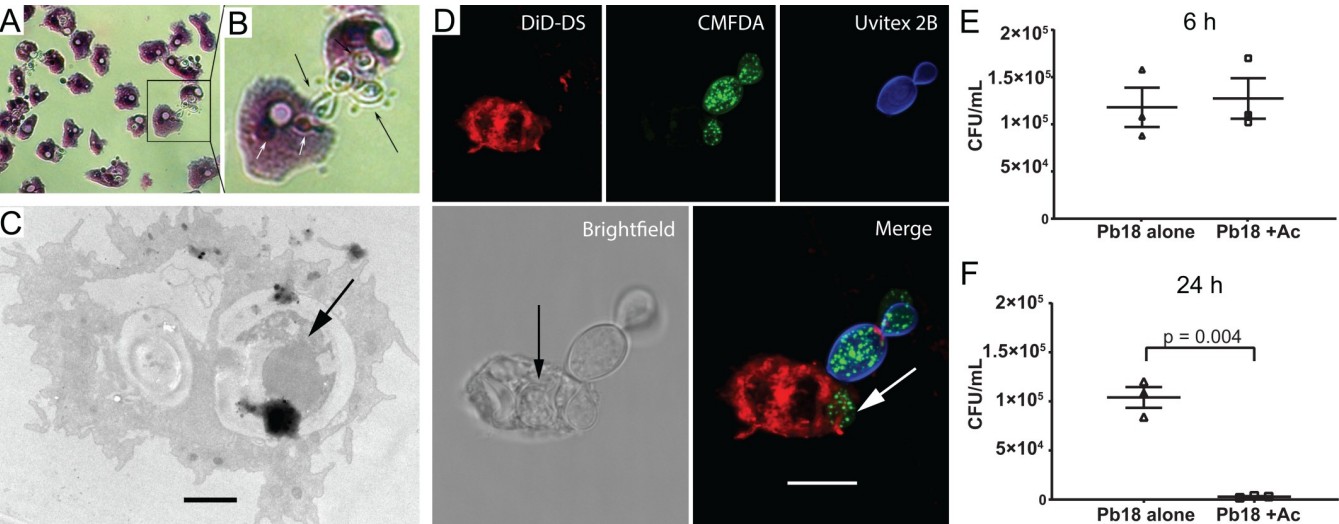

**Fig 4. *P. brasiliensis* Pb18 interaction with an axenic *A. castellanii* strain.** A) *P. brasiliensis* and *A. castellanii* were co-incubated at an MOI of one for one hour at 28˚C, and then stained with Giemsa and observed by light microscopy. B) Enlargement of the area depicted in the square region of panel A. C) TEM of the interaction of *A. castellanii* and Pb18 cells. Incubation was at an MOI of one for six hours at 28˚C and then fixed. The black arrow indicates an internalized *P. brasiliensis* (Scale bar = 2 μm). D) Confocal microscopy. *A. castellanii* was dyed with DiD-DS (red), while *P. brasiliensis* cells were labeled first with CMFDA (green), and after the interaction with Uvitex 2B (blue). The arrows show fungal cells inside an amoeba. (Scale bar = 10 μm). E and F) Survival of *P. brasiliensis* after interaction with *A. castellanii.* Incubation was at an MOI of two at 28˚C for six (E) or 24 hours (F), using the fungus alone as a control. After the interaction amoeba cells were lysed and fungal cells were plated for CFU counting. The figure depicts the results of three independent experiments. The error bars represent the standard error of the mean.

only at 6 hours of interaction. In addition, the armadillo isolate was also able to survive the interaction with amoebae better than the other two strains (Fig 5C). We observed an increase of roughly five-fold in the CFU of T16B1 after the interaction in comparison to the other two strains.

### Sequential interaction of *P. brasiliensis* with *A. castellanii* selects for fungal cells with significant changes in their ability to survive and interact with different host models

We evaluated if sequential rounds of interaction of *P. brasiliensis* with amoebae were able to select fungal cells with increased virulence when compared to the non-passaged strain as previously reported for *H. capsulatum* [7]. We submitted the fungus to six hours of interaction with amoebae at 28˚C in PYG medium. The amoebae were then lysed and all interacting fungal cells (intracellular and extracellular) were collected and plated in solid BHI-sup medium. This procedure was repeated 4 additional times, resulting in Pb18-Ac strains. Both Pb18-Ac and Pb18 strains were used in co-cultures with *A. castellanii* and J774 macrophages and to infect *G. mellonella* and BALB/c mice. When comparing Pb18-Ac cells to the non-passaged strain, the phagocytosis by amoebae decreased from 55.4% to 44.6% (Fig 6A), the proportion of dead amoebae increased from 10.8% to 15.9% (Fig 6B) and the number of fungal CFUs increased 2.5-fold (Fig 6C).

Additionally, we also tested whether the changes in Pb-Ac interaction with amoebae could also be translated into other models. The number of recovered fungi in the wells with Pb18-Ac was significantly higher than the control strain after six hours of interaction with J774 macrophages (Fig 6D). Additionally, Pb18-Ac was also able to kill *G. mellonella* larvae and BALB/c mice significantly faster than the non-passaged strain (Fig 6E and 6F).

### Sequential passaging of *P. brasiliensis* affects the accumulation of selected virulence transcripts and increases the accumulation of cell wall α-glucans

We hypothesized that sequential interactions selected for cells with genetic or epigenetic changes that increased expression of genes that are involved in host-pathogen interaction. Quantitative PCR analysis was then carried out to search for changes in the levels of fungal transcripts that were previously shown to be modulated upon interaction with amoebae or macrophages [28–30]. No major changes were noticed in the accumulation of the transcripts of the selected genes between the two strains. The minor changes observed included a slight increase of *HADH* and *HSP60* in Pb18-Ac (Fig 7B and 7E) and a slight decrease of *MS1* and *SOD1* expression (Fig 7C and 7D).

As we observed a decrease in the percentage of phagocytosis of *A. castellanii* co-incubated with the passaged strain, we evaluated if there were any changes in the fungal surface that might affect its internalization. For that, paraformaldehyde-fixed Pb18 and Pb18-Ac cells were incubated with the monoclonal antibody MOPC-104E, which binds to fungal α-(1,3)- glucans [31], and analyzed by flow cytometry. Fig 7H shows a 3.3-fold increase in the signal for α-(1,3)-glucan in the passaged cells relative to the non-passaged strain. As an increase in the accumulation of α-glucan synthase transcripts was not detected by qPCR (Fig 7A), these results suggest that passaging through amoebae affect the content of α-(1,3)-glucans in *P. brasiliensis* cell wall through mechanisms other than mRNA accumulation.

## Discussion

*Paracoccidioides spp*. are thermodimorphic fungal pathogens that cause PCM, a systemic mycosis prevalent in Latin America [2]. Although this disease has been known for more than a

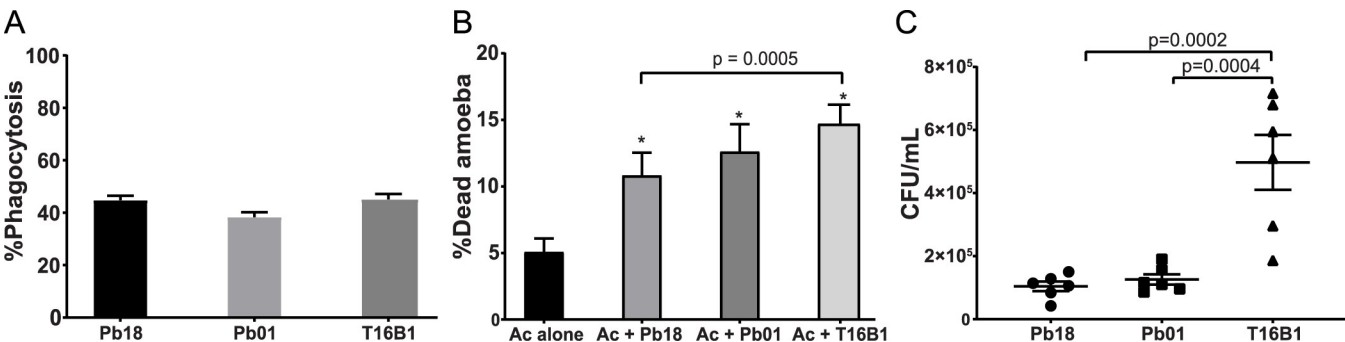

**Fig 5. Interaction of *Paracoccidioides* spp strains with *A. castellanii* at six hours.** Amoebae and three different strains of *Paracoccidioides* spp (Pb18 –*P. brasiliensis*, Pb01 –*P. lutzii*, PbT16B1 –*P. brasiliensis* isolated from an armadillo spleen) were co-incubated at an MOI of two for six hours at 28˚C. A) Percentage of *A. castellanii* cells interacting with *Paracoccidioides* spp. The interaction was assessed by counting at least 100 phagocytes cells per replicate of each sample after Giemsa staining of the samples. The bars represent 95% confidence intervals. B) Viability of *A. castellanii* upon interaction with *Paracoccidioides* spp. The viability was assessed by counting at least 100 cells per replicate of each sample after staining with trypan blue. The bars represent means plus 95% confidence intervals. C) Survival of fungal cells from different strains of *Paracoccidioides* spp following interaction with amoebae. The error bars represent standard error of the mean. Figures depict the combined results of at least three independent experiments. *All the strains showed a significant difference in the % of dead amoebae at six hours relative to the control amoebae growing alone.

century, there are still many unsolved questions about the ecology of its agents. Direct isolation of *Paracoccidiodes* spp. from soil is challenging and has been reported only a few times.

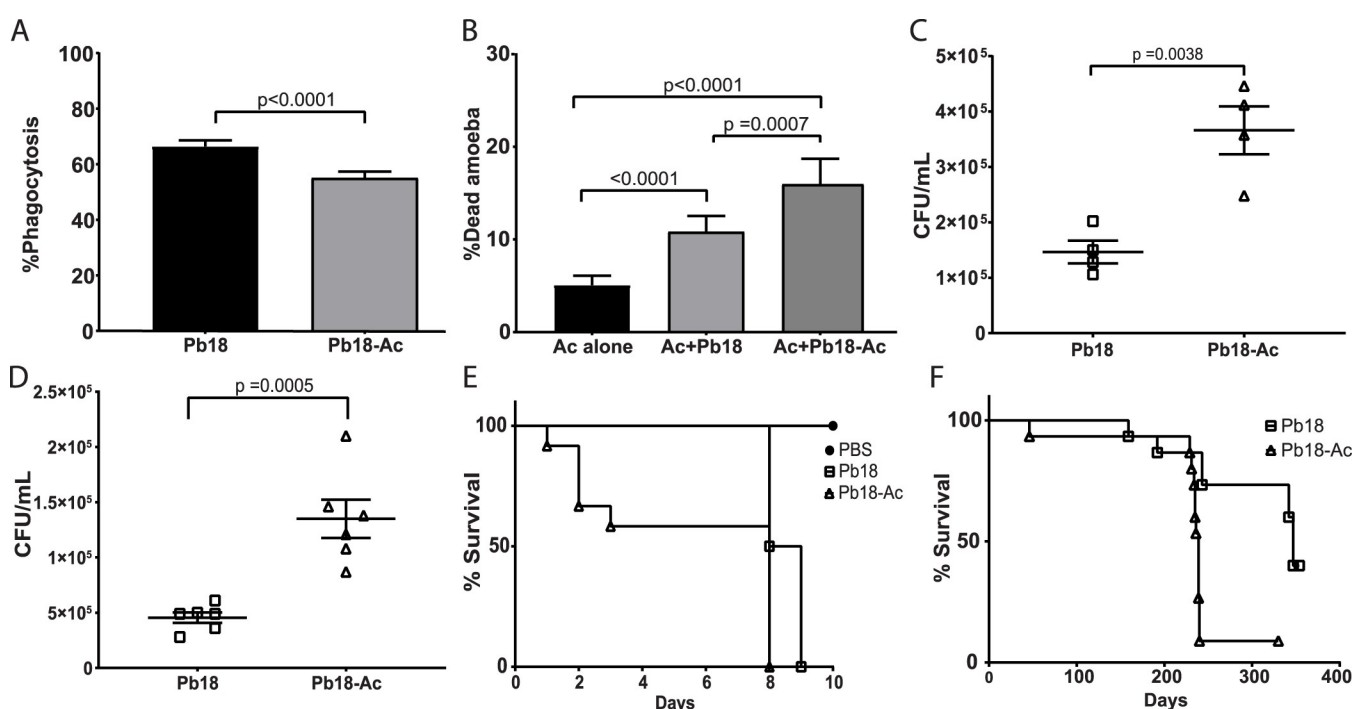

**Fig 6. Effects of sequential passaging of Pb18 with amoebae, assessed in several models of infection.** Pb18 and Pb18 Pb-Ac cells were co-incubated with *A. castellanii* at an MOI of two at 28˚C for six hours. A) Percentage of *A. castellanii* cells interacting with Pb18 and Pb18-Ac. B) Viability of *A. castellanii* after six hours of interaction with Pb18 and Pb18-Ac. C) Survival of Pb18 and Pb18-Ac upon interaction with *A. castellanii*. D) Survival of Pb18 and Pb18 Pb-Ac upon interaction with J774 macrophages. E) Survival curve of *G. mellonella* infected with Pb18 and Pb18-Ac. The curve is representative of two biological replicates. P<0.0001 for the comparison of the survival curve of larvae infected with the two different strains (log-rank test). F) Survival curve of BALB/c mice infected with Pb18 or Pb18 Pb-Ac. Each group had 15 mice. p = 0.0003 for the comparison of the survival curve of mice infected with the two strains (log-rank test). A-D depict the combined results of at least two independent experiments. The bars represent means plus 95% confidence intervals in A and B and standard error mean in C and D.

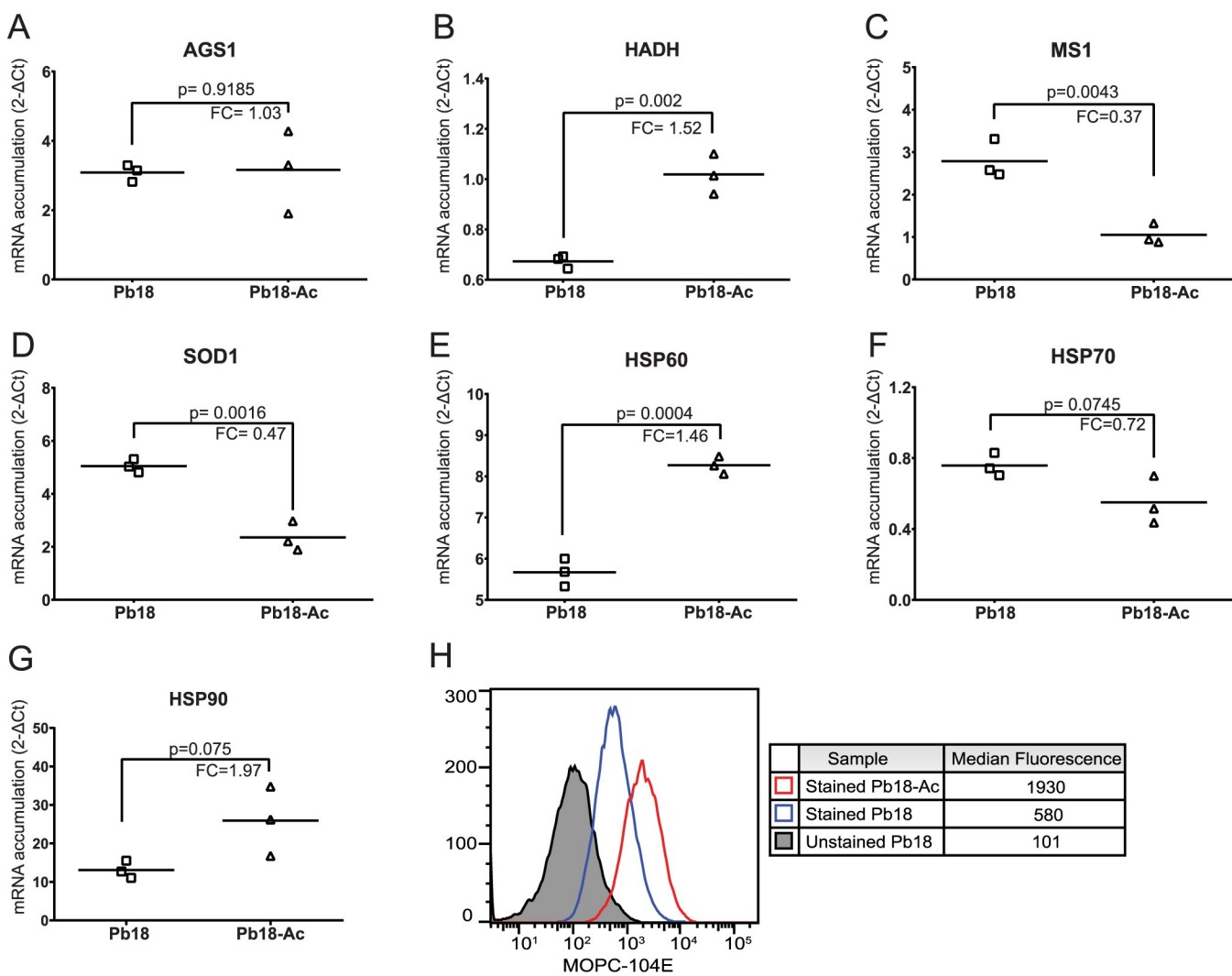

**Fig 7. Changes in *P. brasiliensis* Pb18 gene expression after cycles of interaction with amoeba.** A–G) Modulation of Pb18 gene expression after previous passages with amoebae. Transcript accumulation was determined by the comparative threshold method using the ΔCt value obtained after normalization with the constitutively expressed gene L34. Data are reported as individual $2^{-\Delta\Delta Ct}$ values of three independent experiments for each group and the bar represents their respective means. FC = fold change in mRNA accumulation, obtained as the ratio Pb18-Ac/Pb18. A) AGS1: α-glucan synthase, B) HADH: Hydroxyacyl-CoA Dehydrogenase, C) MS1: malate synthase, D) SOD1: superoxide dismutase 1, E) HSP60: Heat shock protein 60, F) HSP70: Heat shock protein 70, G) HSP90: Heat shock protein 90. H) Cell surface staining of α glucan in the surface of Pb18 cells that were submitted (Pb18-Ac) or not (Pb18) to serial passages of interaction with *A. castellanii*. Cells from the two cell types (Pb18 and Pb18-Ac) were labeled with the antibody to α-glucan MOPC 104E and then with secondary IgM-Alexa fluor 488 antibody and analyzed by flow cytometry.

However, detection of *Paracoccidioides* DNA from soil and aerosol samples is much more widely reported, suggesting that these fungi are saprophytes like those in the genera *Cryptococcus*, *Histoplasma* and *Blastomyces* [14, 32, 33].

In this study, we analyzed the interaction of *Paracoccidioides* spp. with different soil amoebae. Our underlying hypothesis was that the *Paracoccidioides* spp. virulence traits could have been selected by interactions with environmental predators such as amoebae, as previously proposed for other soil-borne fungal and bacterial pathogens. We performed interaction assays between *Paracoccidioides* spp cells and four different amoebae, including three amoebae that we isolated from soil—*Acanthamoeba* spp, *A. spelaea*, *V. vermiformis*—and an axenic laboratory strain of *A. castellanii*. *Acanthamoeba* is a genus of soil amoeba that can cause keratitis

and granulomatous amoebic encephalitis [34]. *A. spelaea* was first identified in 2009 and can also be involved in human keratitis, but there is little information about it in the literature [35]. *V. vermiformis* is frequently isolated from soil and water environments, including hospital tap water [36]. Interestingly, all three genera have previously been reported to harbor potentially pathogenic intracellular microbes such as *Legionella pneumophila* [37–39]. Additionally, there are several reports on the interaction of *A. castellanii* with different pathogenic fungi, and *V. vermiformis* has been shown to promote *Candida* spp. growth in tap water and conidial filamentation of *A. fumigatus* [7, 8, 10, 40–42]. In our experiments, all four amoebae were able to internalize and kill *Paracoccidioides* cells; furthermore, non-axenic amoeba cultures were able to grow using fungal cells as their major food source.

In addition to phagocytosis, the different microscopy approaches we used revealed dead fungal cells with severely altered morphology, including perforations in their surface. Along with the fact that a *Paracoccidioides* spp mother cell with multiple buds is much larger than most amoebae we observed, these perforated cells point to other mechanisms of *Paracoccidioides* spp killing by amoebae that do not require phagocytosis. Our observations are supported by previous reports from late 1970s of giant vampyrellid soil amoebae that perforated conidia and hyphae of soil fungi such as *Cochliobolus sativus* and *Alternaria alternata* to feed upon their contents [43, 44]. More recently, Radosa and collaborators also described fungivorous amoebae feeding on filamentous fungi by perforating and invading hyphae [45]. This strategy bears a striking resemblance to the direct antimicrobial actions of vertebrate CD8+ T and NK cells, which directly interact with and kill *Cryptococcus neoformans* and *P. brasiliensis* by perforating their cell walls [46–48]. The selective pressure put on fungi by this amoeba feeding strategy could have led to fungi with cell walls that are more resistant to the antimicrobial actions of CD8+ T and NK cells.

The effective predation of *Paracoccidioides* spp. cells by amoebae we observed in our experiments resembles observations made several decades ago of *A. castellanii* using *C. neoformans* as a food source and playing a role in controlling fungi in the environment [49, 50]. In contrast, a number of studies published in the early 2000's showed pathogenic fungi such as *C. neoformans*, *Histoplasma capsulatum*, and *Blastomyces dermatitidis* surviving the interaction with amoebae [7, 8]. This apparent contradiction might be explained by a fundamental difference in experimental design. Fu and Casadevall have recently reported that divalent cations increase the survival of amoebae that are interacting with *C. neoformans* and also potentiate their antifungal activity [51]. We performed our interaction experiments in PYG medium or PAS, both of which contain calcium and magnesium, whereas the reports in which fungi survived the interaction were made in PBS without divalent cations.

The apparently higher susceptibility of *Paracoccidioides* to amoeba in our experiments might also be explained by the use of the fungal cells which were not regularly passaged through an animal host. This seemed to be the case of the cultures we used, considering the long median survival of mice infected with the non-passaged strain. Although *Paracoccidioides* spp is a primary pathogen, previous reports and our own experience have shown that prolonged in vitro subculture of these fungi leads to attenuation or loss of virulence, which can be restored by animal passaging [52, 53]. However, when we compare the interaction of the same fungal strain with amoeba and macrophages, Pb18 was still able to kill a high proportion of macrophages. These data suggest that despite the several similarities between amoebae and macrophages, there are important differences between these two cell host systems and/or that prolonged in vitro subculturing caused the fungal strain to lose virulence attributes that are more specific for its interaction with amoebae.

We further analyzed the interaction of *A. castellanii* with the sister species *P. lutzii* (Pb01) and with a *P. brasiliensis* strain isolated from an armadillo (PbT16B1). Regarding rates of

internalization, the ability to kill amoebae and to survive the interaction, Pb01 behaved identically to Pb18. However, the interaction of *A. castellanii* with T16B1 revealed that this strain was more efficient in surviving and killing the amoebae. Given that the armadillo strain was isolated about 7 years ago whereas Pb18 and Pb01 were isolated about 90 and 30 years ago, respectively [54, 55], these results point to the attenuation of *Paracoccidioides* spp. after prolonged in vitro subculturing. Our results are also compatible with previous work from other groups showing that *P. brasiliensis* armadillo isolates can be more virulent to mice and hamster models than some clinical strains submitted to prolonged in vitro culturing such as Pb18 [52, 56, 57].

We wanted to evaluate if cycles of interaction with amoebae could revert this apparent virulence attenuation of *Paracoccidioides* spp. Our initial idea was to use the isolated soil amoebae for this assay because that would give us a model more closely related to what might be happening in nature. However, the presence of antibiotic-resistant contaminant bacteria in the isolated amoeba cultures impaired the recovery of fungal cells after the interaction. To overcome this limitation, we relied instead on an axenized laboratory strain of *A. castellanii* that had been validated with several other fungal pathogens [7, 8, 10] as a host to test this hypothesis. Despite being adapted to axenic growth in laboratory conditions, this amoeba was able to efficiently ingest and kill *P. brasiliensis* cells. Sequential cycles of fungal interaction with amoebae, each cycle short enough to allow fungal survival, selected for changes in the virulence of *P. brasiliensis* Pb18. In new interaction assays with *A. castellanii*, passaged strains (Pb18-Ac) were more efficient in evading phagocytosis, surviving the interaction and killing the amoebae than the non-passaged strains (Pb18). The passaged strain was also able to survive better the interaction with J774 macrophages and had an increased ability to kill *G. mellonella* larvae and mice, confirming that interaction with *A. castellanii* was able to select for broader changes in fungal virulence. These results are in accordance with what was described for *H. capsulatum* and *C. neoformans* upon their interaction with amoebae [7, 58]. Given that Pb18-Ac cells evaded phagocytosis more effectively and that cell wall α-(1,3) glucan has been shown to mask recognition of dimorphic fungi by host cells [31], we hypothesized that the passaged strain could have an increase in cell wall α-(1,3) glucans relative to Pb18. Flow cytometry experiments confirmed this hypothesis, suggesting a similar role for α-(1,3) glucans in avoiding phagocytic receptors in amoebae and mammals. Interestingly, this increase in cell wall α-(1,3) glucans was not the result of increased accumulation of the α-glucan synthase transcript, suggesting a non-transcriptional mechanism of regulation. Similarly, most other genes whose expression we tested were not altered or had only small changes in expression in the passaged strain.

Overall, our results fit into the recently formulated amoeboid predator-fungal animal virulence hypothesis whereby there is a nexus of causation from selective pressure of amoeboid environmental predators and the evolution of fungal virulence against mammals [59]. We have shown that *Paracoccidioides* spp. may indeed interact with different amoebae species in its environment, and that soil protozoans, among many other predators, could have a role as a selective pressure for the emergence virulence traits in this genus. Given that PCM is not usually transmitted from a person to another, the better understanding of the environment in which this fungus lives could lead to improved preventative measures aiming at decreasing the exposure of rural workers to the fungus.

## Supporting information

**S1 Fig. Interaction of *P. brasiliensis* Pb18 cells with soil amoeba isolates in solid plates of non-nutrient agar.** A suspension of $1.5 \times 10^7$ *P. brasiliensis* Pb18 cells was plated onto non-nutrient agar and spotted with $10^4$ cells of *Acanthamoeba* spp (panels A-D), A. *spelaea* (panels

E-H) or *V. vermiformis* (panels I-L) in 10-microlitre aliquots directly in the middle of the fungal cell lawn. The plates were incubated at 25°C for 19 days and inspected for the formation of lysis plates and fungal cell digestion at day 1 (panels A, E, I), day 3 (panels B, F, J), day 7 (panels C, G, K) and day 19 of interaction (panels D, H, L). Black arrowheads depict regions of fungal clearance.
(TIF)

**S2 Fig. Morphology of colonies of *P. brasiliensis* Pb18 cells after 30 days of interaction with soil amoeba.** Five microliters of a suspension of 1.5 x10$^7$ *P. brasiliensis* Pb18 cells were plated onto non-nutrient agar, spotted with amoeba isolates and photographed after 30 d of interaction. A) Control colony of *P. brasiliensis* Pb18 displaying intense filamentation. B) Colony of *P. brasiliensis* Pb18 co-incubated with A. *spelaea*. C) Colony of *P. brasiliensis* Pb18 co-incubated with *V. vermiformis*. D) Colony of *P. brasiliensis* Pb18 co-incubated with *Acanthamoeba* spp.
(TIF)

**S3 Fig. Changes in *P. brasiliensis* Pb18 cell morphology after interaction with different soil amoebae.** The isolates were co-incubated with Pb18 at an MOI of two at 25°C for 4 or 24 hours in PAS and fixed for TEM or SEM. A-B) TEM of *P. brasiliensis* cells growing alone. Scale bars = 10 μm. E, F) TEM showing the morphology of *P. brasiliensis* cells after the interaction with *Acanthamoeba* spp. Scale bars = 5 μm. I and J) TEM showing the morphology of *P. brasiliensis* cells after the interaction with *A. spelaea*, or *V. vermiformis*, respectively. Scale bars = 5 μm and 500 nm. C-D) SEM of *P. brasiliensis* cells growing alone. Scale bars = 5 μm and 10 μm, respectively. G and H) SEM showing the morphology of *P. brasiliensis* cells after the interaction with *V. vermiformis* or *A. spelaea*. Scale bars = 10 μm and 5 μm. K, L) SEM showing the morphology of *P. brasiliensis* cells after the interaction with *V. vermiformis* or *Acanthamoeba* spp. respectively. Scale bars = 10 μm and 5 μm, respectively. Red arrowheads indicate fungal cells, or their remains.
(TIF)

**S4 Fig. Kinetics of phagocytosis of *P. brasiliensis* by *A. castellanii*.** Amoebae and *P. brasiliensis* yeast cells (CMFDA labeled) were co-incubated (MOI of two). At each time point, the percentage of phagocytosis was evaluated by fluorescence microscopy. A minimum of 300 amoebae per sample was analyzed to calculate the percentage of phagocytosis. The plot represents the results from three independent experiments each performed in triplicate. The error bars represent the 95% confidence interval.
(TIF)

**S5 Fig. Viability of *A. castellanii* and J774 macrophages after interaction with *P. brasiliensis* Pb18.** A) Amoeba cells were incubated alone or in the presence of *P. brasiliensis* at 28°C or 37°C for six, 24 and 48 hours (MOI of two). (B) J774 macrophages were incubated alone or in the presence of *P. brasiliensis* at 37°C in a $CO_2$ incubator for six, 24 and 48 hours (MOI of two). Viability was assessed at each time point by counting at least 300 phagocytes cells per replicate after staining with trypan blue. The error bars indicate the 95% confidence interval.
(TIF)

**S6 Fig. Viability of *A. castellanii* (Ac) after interaction with different *Paracoccidioides* spp strains.** A) Amoebae were incubated alone or in the presence of *P. brasiliensis* Pb18, *P. lutzii* Pb01 or *P. brasiliensis* T16B1 yeast cells at 28°C for six, 24 and 48 hours (MOI of two). Viability was assessed at each time point by counting at least 300 phagocytes cells per replicate after

staining with trypan blue. The error bars indicate the 95% confidence interval.
(TIF)

**S1 Table. Genes and primer sequences for qPCR experiments.**
(DOCX)

## Acknowledgments

We are grateful for the valuable help of several people along the development of this work including Barbara Smith, Thales D. Arantes, Raquel Theodoro, Marluce F. Hrycyk, Carlos Eduardo Winter, Jessica Ferrão, Gabriela Matos, Cristine Barreto, Izabella Monteiro Rizzi de Azevedo, Bianca Oliveira do Vale Lira, and Calliandra de Souza.

## Author Contributions

**Conceptualization:** Patrícia Albuquerque, Allan Jefferson Guimarães, Eduardo Bagagli, Maria Sueli Soares Felipe, Arturo Casadevall, Ildinete Silva-Pereira.

**Formal analysis:** Patrícia Albuquerque, André Moraes Nicola, Diogo Almeida Gomes Magnabosco, Lorena da Silveira Derengowski, Luana Soares Crisóstomo, Luciano Costa Gomes Xavier, Stefânia de Oliveira Frazão, Fernanda Guilhelmelli, Marco Antônio de Oliveira, Fabián Andrés Hurtado, Marcus de Melo Teixeira, Hugo Costa Paes.

**Funding acquisition:** Patrícia Albuquerque, Arturo Casadevall, Ildinete Silva-Pereira.

**Investigation:** Patrícia Albuquerque, André Moraes Nicola, Diogo Almeida Gomes Magnabosco, Lorena da Silveira Derengowski, Luana Soares Crisóstomo, Luciano Costa Gomes Xavier, Stefânia de Oliveira Frazão, Fernanda Guilhelmelli, Jhones do Nascimento Dias, Hugo Costa Paes.

**Methodology:** Patrícia Albuquerque, André Moraes Nicola.

**Resources:** Eduardo Bagagli, Maria Sueli Soares Felipe, Arturo Casadevall, Ildinete Silva-Pereira.

**Supervision:** Patrícia Albuquerque, Lorena da Silveira Derengowski.

**Writing – original draft:** Patrícia Albuquerque.

**Writing – review & editing:** Patrícia Albuquerque, André Moraes Nicola, Marcus de Melo Teixeira, Allan Jefferson Guimarães, Hugo Costa Paes, Arturo Casadevall, Ildinete Silva-Pereira.

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
