## [Decision Letter · Decision Letter 0]

6 Aug 2019

Dear Dr Albuquerque:

Thank you very much for submitting your manuscript "A hidden battle in the dirt: soil amoebae interactions with Paracoccidioides spp" (PNTD-D-19-00966) for review by PLOS Neglected Tropical Diseases. Your manuscript was fully evaluated at the editorial level and by independent peer reviewers. The reviewers appreciated the attention to an important topic but identified some aspects of the manuscript that should be improved.

We therefore ask you to modify the manuscript according to the review recommendations before we can consider your manuscript for acceptance. Your revisions should address the specific points made by each reviewer.

(1) A letter containing a detailed list of your responses to the review comments and a description of the changes you have made in the manuscript.

(2) Two versions of the manuscript: one with either highlights or tracked changes denoting where the text has been changed (uploaded as a "Revised Article with Changes Highlighted" file ); the other a clean version (uploaded as the article file).

(3) If available, a striking still image (a new image if one is available or an existing one from within your manuscript). If your manuscript is accepted for publication, this image may be featured on our website. Images should ideally be high resolution, eye-catching, single panel images; where one is available, please use 'add file' at the time of resubmission and select 'striking image' as the file type. 

Please provide a short caption, including credits, uploaded as a separate "Other" file. If your image is from someone other than yourself, please ensure that the artist has read and agreed to the terms and conditions of the Creative Commons Attribution License at http://journals.plos.org/plosntds/s/content-license (NOTE: we cannot publish copyrighted images). 

(4) Appropriate Figure Files 

Please remove all name and figure # text from your figure files upon submitting your revision. Please also take this time to check that your figures are of high resolution, which will improve both the editorial review process and help expedite your manuscript's publication should it be accepted. Please note that figures must have been originally created at 300dpi or higher. Do not manually increase the resolution of your files. For instructions on how to properly obtain high quality images, please review our Figure Guidelines, with examples at: http://journals.plos.org/plosntds/s/figures

While revising your submission, please upload your figure files to the Preflight Analysis and Conversion Engine (PACE) digital diagnostic tool, https://pacev2.apexcovantage.com/ PACE helps ensure that figures meet PLOS requirements. To use PACE, you must first register as a user. Then, login and navigate to the UPLOAD tab, where you will find detailed instructions on how to use the tool. If you encounter any issues or have any questions when using PACE, please email us at figures@plos.org.

We hope to receive your revised manuscript by Oct 05 2019 11:59PM. If you anticipate any delay in its return, we ask that you let us know the expected resubmission date by replying to this email.

To submit your revised files, please log in to https://www.editorialmanager.com/pntd/

Sincerely,

Todd B. Reynolds

Deputy Editor

Todd Reynolds

Deputy Editor

Reviewer's Responses to Questions

**Key Review Criteria Required for Acceptance?**

**Methods**

-Are the objectives of the study clearly articulated with a clear testable hypothesis stated?

-Is the study design appropriate to address the stated objectives?

-Is the population clearly described and appropriate for the hypothesis being tested?

-Is the sample size sufficient to ensure adequate power to address the hypothesis being tested?

-Were correct statistical analysis used to support conclusions?

-Are there concerns about ethical or regulatory requirements being met?

Reviewer #1: The methods are clear, concise and well designed to address the study objective. The sample size was enough to ensure the results were statistically valid. Ethical requirements were met.

It would be good if the authors give references for all their methods i.e. electron microscopy (EM) work, etc.

For EM, it would have been good to include a control of cryptococcal cells alone to exclude the harsh preparatory steps of EM as the reason for collapsed cells.

Reviewer #2: Yes.

Reviewer #3: The authors should not include results in the Introduction section "These studies showed that amoebae efficiently ingest and kill Paracoccidioides spp. yeast cells, and that this interaction selects for fungi with increased virulence. Our results support the hypothesis that interaction with neighbor soil predators selects for traits that allow survival of Paracoccidioides spp. in mammalian hosts and add to the existing evidence for the amoeboid predator-animal virulence hypothesis [13].".

The objectives of the study are clearly articulated

The methods were described with sufficient detail and there is information to replicate.

**Results**

-Does the analysis presented match the analysis plan?

-Are the results clearly and completely presented?

-Are the figures (Tables, Images) of sufficient quality for clarity?

Reviewer #1: The analysis match the analysis plan. However, lack of a control for EM work questions the findings. 

To harvest the cryptococcal cells, amoeba had to be lysed. Could the latter see amoeba also contributing to the observed levels of glucans?

How did the authors normalise the mRNA levels of genes of interest?

Reviewer #2: Yes.

Reviewer #3: The results are clearly and completely presented, and the figures are sufficient quality.

**Conclusions**

-Are the conclusions supported by the data presented?

-Are the limitations of analysis clearly described?

-Do the authors discuss how these data can be helpful to advance our understanding of the topic under study?

-Is public health relevance addressed?

Reviewer #1: The public health relevance is adequately addressed. The conclusions that reached confirm what is already known in literature.

Reviewer #2: Could be modified.

Reviewer #3: The limitations of study were not described. I suggest Include and discuss the limitations of your study.

**Editorial and Data Presentation Modifications?**

Reviewer #1: Minor revision

Reviewer #2: Lines 186-187: the original strain is part of the comparison, so not a control. Could be “with strains Pb18-Ac or the non-passaged form Pb18.”

Line 203: space “glucan, and”.

Line 206: “analyses” plural.

Line 234: “hits at GenBank”.

Line 274: “were” [bacteria plural].

Line 354: “through mechanisms other than mRNA accumulation”.

Line 386: “1970s”.

References: check the formatting is correct, as the journal will be unlikely to do so, e.g. italics on species or gene names.

Lines 728-729: change to be “Figure 7- Changes in P. brasiliensis Pb18 gene expression after cycles of interaction with amoeba. Transcript accumulation…” the first sentence in bold.

Reviewer #3: “Minor Revision”

**Summary and General Comments**

Reviewer #1: The presented speak to the idea that amoebal predatory pressure may select Paracoccidioides to become pathogenic. This (selection) has also been observed elsewhere for other fungal species. Therefore, there is no novelty in the work. Nonetheless, the work still highlights the medical importance of the fungus and acquisition of its virulence. 

The authors should consider sufficiently addressing the issues that are raised.

Reviewer #2: The manuscript reports on the interaction of Paracoccidioides strains with three amoeba species isolated from armadillo burrows, which are associated with Paracoccidioides. Passage of fungal isolates through amoeba lead to ‘evolved’ strains with increased virulence in both mice and an insect model, Galleria mellonella, leading to support of the hypothesis that amoeba (or other predators) are important in maintaining the virulence of Paracoccidioides when it encounters humans.

This is a substantial body of work. One reflection of this is that mouse studies went for 12 months. The electron microscopy adds the visual appeal of the work, and the level of replication to infer statistical significance is noteworthy. The work will appeal to the medical mycology community, especially those working on pathogens that have environmental reservoirs. Probably the main point of concern is how the results are interpreted overall.

1) ‘in controlled laboratory conditions using mostly pure and axenic cultures of

laboratory-adapted predators; this very informative system is nonetheless an extreme

simplification of the complex ecosystem soil saprophytes find in nature’ is a very valid point, and it is relevant that here new amoeba strains were isolated and characterized for their interaction with Paracoccidioides strains. However, counter to this the authors then rely on the standard A. castellanii strain ATCC 30234 [which was identified as a natural predator of another pathogenic fungus, Cryptococcus neoformans, so not as artificial as other amoeba strains] for subsequent characterization of the impact of amoeba on Paracoccidioides by passaging.

2) ‘Sequential co-cultivation of Paracoccidioides with A. castellanii selected for

phenotypical traits related to the survival of the fungus within a natural predator as well as in

murine macrophages and in vivo (Galleria mellonella and mice). This increase in virulence…’ However; Pb18 is already an attenuated strain, so while certainly the passaging has changed the virulence properties, it is unclear that this is not just back to the baseline level.

3) A point that needs addressing more explicitly, counter to statements like “the higher susceptibility of Paracoccidioides to amoeba”, is that some survive predation to become more fit in causing disease. The data (such as figure 2) show that a considerable proportion of the fungal isolates survive.

Reviewer #3: The manuscript "A hidden battle in the dirt: soil amoebae interactions with Paracoccidioides spp." is well written in an engaging and the level is appropriate to readership of PLoSNTD. I have no hesitation in recommending that it be accepted for publication after a few minor details have been attended to, without the need for further experimental work.

PLOS authors have the option to publish the peer review history of their article (what does this mean?). If published, this will include your full peer review and any attached files.

Reviewer #1: No

Reviewer #2: No

Reviewer #3: No

---

## [Editor Report · Decision Letter 1]

2 Sep 2019

[EXSCINDED]

Dear Dr Albuquerque,

We are pleased to inform you that your manuscript, "A hidden battle in the dirt: soil amoebae interactions with Paracoccidioides spp", has been editorially accepted for publication at PLOS Neglected Tropical Diseases.

Before your manuscript can be formally accepted and sent to production you will need to complete our formatting changes, which you will receive in a follow up email. Please note: your manuscript will not be scheduled for publication until you have made the required changes.

IMPORTANT NOTES

* Copyediting and Author Proofs: To ensure prompt publication, your manuscript will NOT be subject to detailed copyediting and you will NOT receive a typeset proof for review. The corresponding author will have one final opportunity to correct any errors when sent the requests mentioned above. Please review this version of your manuscript for any errors.

* If you or your institution will be preparing press materials for this manuscript, please inform our press team in advance at plosntds@plos.org. If you need to know your paper's publication date for media purposes, you must coordinate with our press team, and your manuscript will remain under a strict press embargo until the publication date and time. PLOS NTDs may choose to issue a press release for your article. If there is anything that the journal should know, please get in touch.

*Now that your manuscript has been provisionally accepted, please log into EM and update your profile. Go to http://www.editorialmanager.com/pntd, log in, and click on the "Update My Information" link at the top of the page. Please update your user information to ensure an efficient production and billing process.

*Note to LaTeX users only - Our staff will ask you to upload a TEX file in addition to the PDF before the paper can be sent to typesetting, so please carefully review our Latex Guidelines [http://www.plosntds.org/static/latexGuidelines.action] in the meantime.

Best regards,

Todd B. Reynolds

Deputy Editor

Todd Reynolds

Deputy Editor

p.p1 {margin: 0.0px 0.0px 0.0px 0.0px; line-height: 16.0px; font: 14.0px Arial; color: #323333; -webkit-text-stroke: #323333}span.s1 {font-kerning: none}

---

## [Editor Report · Acceptance letter]

30 Sep 2019

Dear Dr Albuquerque,

We are delighted to inform you that your manuscript, "A hidden battle in the dirt: soil amoebae interactions with Paracoccidioides spp," has been formally accepted for publication in PLOS Neglected Tropical Diseases.

Best regards,

Serap Aksoy

Editor-in-Chief

Shaden Kamhawi

Editor-in-Chief
